# Assessment of the performance of nonfouling polymer hydrogels utilizing citizen scientists

**Niko Hansen, Adriana Bryant, Roslyn McCormack, Hannah Johnson, Travis Lindsay, Kael Stelck, Matthew T. Bernards** *

Department of Chemical and Biological Engineering, University of Idaho, Moscow, Idaho, United States of America

* mbernards@uidaho.edu

## Abstract

To facilitate longer duration space travel, flight crew sickness and disease transmission amongst the crew must be eliminated. High contact surfaces within space vehicles provide an opportunity for bacterial adhesion, which can lead to biofilm formation or disease transmission. This study evaluates the performance of several nonfouling polymers using citizen science, to identify the best performing chemistry for future applications as bacteria resistant coatings. The specific polymer chemistries tested were zwitterionic sulfobetaine methacrylate (SBMA), and polyampholytes composed of [2-(acryloyloxy)ethyl] trimethylammonium chloride and 2-carboxyethyl acrylate (TMA/CAA), or TMA and 3-sulfopropyl methacrylate (TMA/SA). Each polymer chemistry is known to exhibit bacteria resistance, and this study provides a direct side-by-side comparison between the chemistries using a citizen science approach. Nearly 100 citizen scientists returned results comparing the performance of these polymers over repeat exposure to bacteria and 30 total days of growth. The results demonstrate that TMA/CAA polyampholyte hydrogels show the best long-term resistance to bacteria adhesion.

**Data Availability Statement:** The data underlying the results presented in this study are available through Dryad: https://doi.org/10.5061/dryad.5dv41ns79.

## Introduction

The potential for significant impact from microbes during long duration space flights is a concern that must be managed to prevent unwanted infection and disease. Microorganisms will always be present aboard manned spacecraft despite stringent sterilization efforts due to humans carrying their own microbiome [1]. Of particular interest is the minimization of biofilm formation on surfaces that are frequently touched by members aboard the International Space Station (ISS). Once formed, even if disinfected, biofilms condition surfaces for future colonization by microbial communities [2, 3] and often require more disinfectant to treat than planktonic cultures [4–6]. Therefore, minimizing the potential formation of biofilms provides another mitigation strategy to ensure a safe environment aboard the ISS.

Due to the impact of infectious disease on Apollo missions 9–13, a large emphasis has been placed on the flight crew Health Stabilization Program (HSP) which has resulted in a considerable reduction in the risk for infectious disease both before and during flight [7]. However,

**Funding:** This research was funded by the National Aeronautics and Space Administration (NASA) as part of the Student Payload Opportunity with Citizen Science (SPOCS) program under Federal Award No 80NSSC17M0021, as a subaward to the University of Idaho. The funders had no role in study design, data collection and analysis, decision to publish, or preparation of the manuscript.

**Competing interests:** The authors have declared that no competing interests exist.

despite the benefits of the HSP, multiple bacteria strains have been isolated from surfaces onboard the ISS [8]. Therefore, implementing additional methods for reducing bacterial adhesion will further reduce the risk of infection and facilitate successful future long-duration missions. Bacteria resistant polymer coatings is one potential approach.

The polymers of interest are nonfouling materials, which are defined in the literature as materials that have less than 5 ng/cm$^2$ of nonspecific protein adsorption, even upon exposure to 100% plasma, serum, or blood [9]. Materials that have achieved this criterion have also demonstrated resistance to bacteria adhesion. One of the most promising families of nonfouling polymers are zwitterionic and mixed charge polymer systems. Zwitterionic polymers are defined as having an equimolar number of anion and cation groups within each monomer subunit [10]. A related family are polyampholyte polymers, whose overall neutral, mixed charge comes from multiple charged monomer subunits [11]. In this study, the potential zwitterionic and polyampholyte systems that are being evaluated are restricted to those that are readily and inexpensively available via commercial sources. Therefore, the evaluation is limited to zwitterionic sulfobetaine methacrylate (SBMA), and polyampholyte mixtures of [2-(acryloyloxy)ethyl] trimethylammonium chloride and 2-carboxyethyl acrylate (TMA/CAA), and TMA and 3-sulfopropyl methacrylate (TMA/SA). This restriction is due to the need to produce many samples to facilitate the citizen scientist evaluation.

SBMA has previously been shown to greatly reduce non-specific protein adsorption and short-term bacteria adhesion [12]. SBMA has also been shown to reduce bacterial adhesion in longer-term (9 days) applications [13]. The TMA/CAA polyampholyte polymer has been shown to perform as a nonfouling material with low nonspecific protein adsorption from full serum [14]. Additionally, TMA/CAA polymers have demonstrated resistance to *Staphylococcus epidermidis* in a pH dependent manner [15]. Finally, TMA/SA polyampholyte systems have similar nonfouling behaviors [16, 17], although their bacteria resistance has not been evaluated previously.

Citizen science is a form of research collaboration that taps into a broad voluntary community of nonprofessionals. This approach has become widely popular in environmental and ecological fields of research [18], and it has seen growing interest for biomedical related research [19]. The advantage of using citizen scientists is the ability to collect larger data sets, which is especially beneficial for biological studies that have large variability. Further, it has been suggested that a significant portion of citizen science investigations produce results that correlate well with more controlled studies, if the citizen science tasks are appropriately designed [20].

In this investigation, citizen scientists in grades 3–5 were used to compare the bacteria resistant performance of SBMA, TMA/CAA, and TMA/SA polymer hydrogels. The objective was to rank these three known nonfouling polymers to identify which chemistry demonstrated the best bacteria resistance in a randomized trial based upon the citizen scientist's observed surface coverage of bacteria. Upon return of nearly 100 sets of results, it was determined that the TMA/CAA polymer hydrogel was consistently ranked as the sample with the least amount of bacteria growth. This suggests that TMA/CAA polymers should be further pursued as bacteria resistant coating for high contact surfaces.

## Results and discussion

In this investigation, citizen scientists were used to collect a significant data set comparing the performance of three nonfouling polymer hydrogels. Large-scale hydrogels were formed in glass baking dishes by scaling up existing hydrogel synthesis procedures [17, 21, 22]. Following polymerization, the hydrogels were swollen to their full equilibrium state by soaking them in 0.075 M NaCl solutions overnight. Finally, the large-scale hydrogels were punched out and

placed into 60 mm petri dishes, and 5 mL of 0.075 M NaCl was added to each dish to keep the samples hydrated before use. Two hundred citizen scientist kits were prepared containing two randomized polymer hydrogels and one agar control.

The citizen scientists were students in grades 3–5, so the experiment was designed with expectations appropriate for this skill level. Upon the receipt of their experiment kit, the citizen scientists were tasked with swabbing a random surface at their home with a provided sterile cotton swab, which was subsequently rubbed on the polymer hydrogels and agar control. Each sample was then given 10 mL of sterile Luria-Bertani nutrient broth. The bacteria exposure was repeated weekly, and the broth was replaced following each subsequent bacteria exposure. Finally, the citizen scientists were asked to monitor their experiment daily, by recording observations and a sketch of the bacteria surface coverage in a provided logbook. An example of the data logbook is shown in Fig 1.

Following 30 days of repeated bacteria exposure and growth, the citizen scientists were asked to identify the samples that had the greatest and least amounts of bacteria present based on the observed overall surface coverage. A total of 91 sets of results were returned by the citizen scientists (45.5% return rate), although one result from Combination 2 was discarded because the citizen scientist did not identify any of their samples as having the least bacteria. Additionally, several classroom sets were set up and photographed, for use by students who made mistakes during their individual experiments. Table 1 provides a summary of the randomized sample combinations, for which data was returned. Fig 2 provides a representative photograph of an experimental trial, with significant bacteria growth visible in one of the three samples.

Fig 3 provides a breakdown of the results collected for each experimental combination summarized in Table 1. In all of the experimental combinations, some citizen scientists identified more than one hydrogel as having the least or most bacteria. These data are presented as either "Mix 1", "Mix 2", or "Mix 3" in Fig 3 to represent combinations of TMA/CAA and TMA/SA,

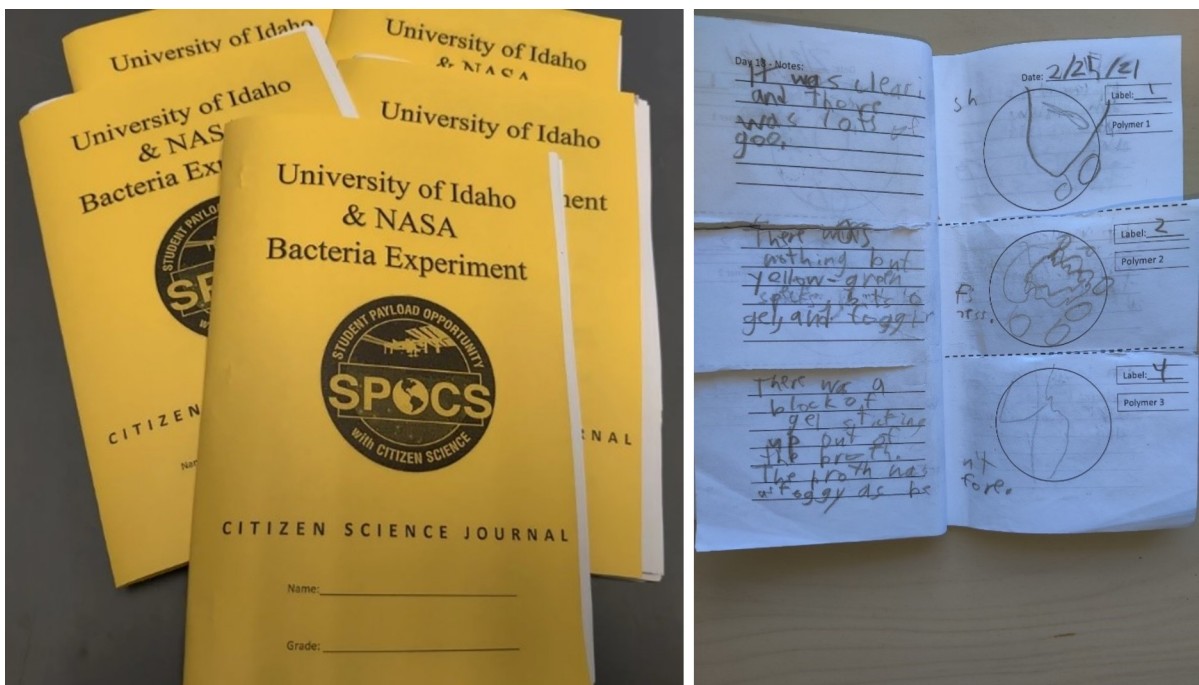

**Fig 1. Citizen scientist logbooks and representative citizen scientist experimental log.**

**Table 1. Summary of the data collected from citizen science kits.**

| Combination 1–34 results |
| :---: |
| TMA/CAA |
| TMA/SA |
| Agar Control |
| Combination 2–31 results |
| TMA/CAA |
| SBMA |
| Agar Control |
| Combination 3–25 results |
| TMA/SA |
| SBMA |
| Agar Control |

TMA/CAA and SBMA, or SBMA and Agar, respectively. In Fig 3, it is clear the TMA/CAA polyampholyte hydrogel was ranked as having the least bacteria more often than the other samples in the two experimental combinations where it was evaluated (Combinations 1 and 2). This is true when the data is presented as both an absolute number of samples (Fig 3A and 3C) or as a percentage of samples evaluated (Fig 3B and 3D). Further, the TMA/CAA polyampholyte hydrogel was never identified as having the most bacteria in any of the results returned by the citizen scientists. In contrast, both the TMA/SA (Combination 1) and SBMA (Combination 2) samples had a subset of results where they were identified as having the most bacteria in these two experimental combinations.

In the third experimental combination, the TMA/SA sample was identified as the sample with the least bacteria, also demonstrating both the highest number and percentage of samples (Fig 3E and 3F). In this subset of the results, TMA/SA was also never identified as having the most bacteria. However, as discussed above, TMA/SA was identified as having the most bacteria in some of the results obtained for Combination 1. Collectively, when the nonfouling samples are compared based on their head-to-head comparisons, it is evident that their nonfouling behaviors are as follows: TMA/CAA > TMA/SA > SBMA.

Fig 4 summarizes the overall citizen scientist evaluations of their individual samples. From the results, it is clear that the TMA/CAA polyampholyte hydrogels outperformed the TMA/SA and SBMA hydrogels. The TMA/CAA hydrogels were ranked as having the least bacteria in 43 of the 65 sets of data that included that chemistry (66.2%). Further, it was not identified as having the most bacteria in any of the data sets. The results also suggest that the TMA/SA hydrogel

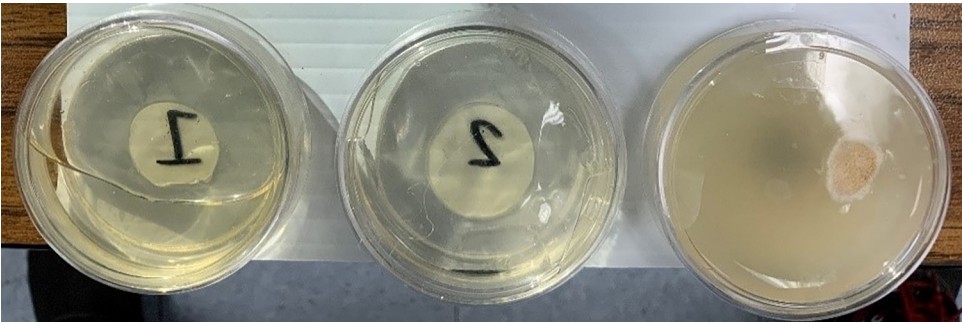

**Fig 2. Representative photograph of an on-going bacteria resistance experiment.**

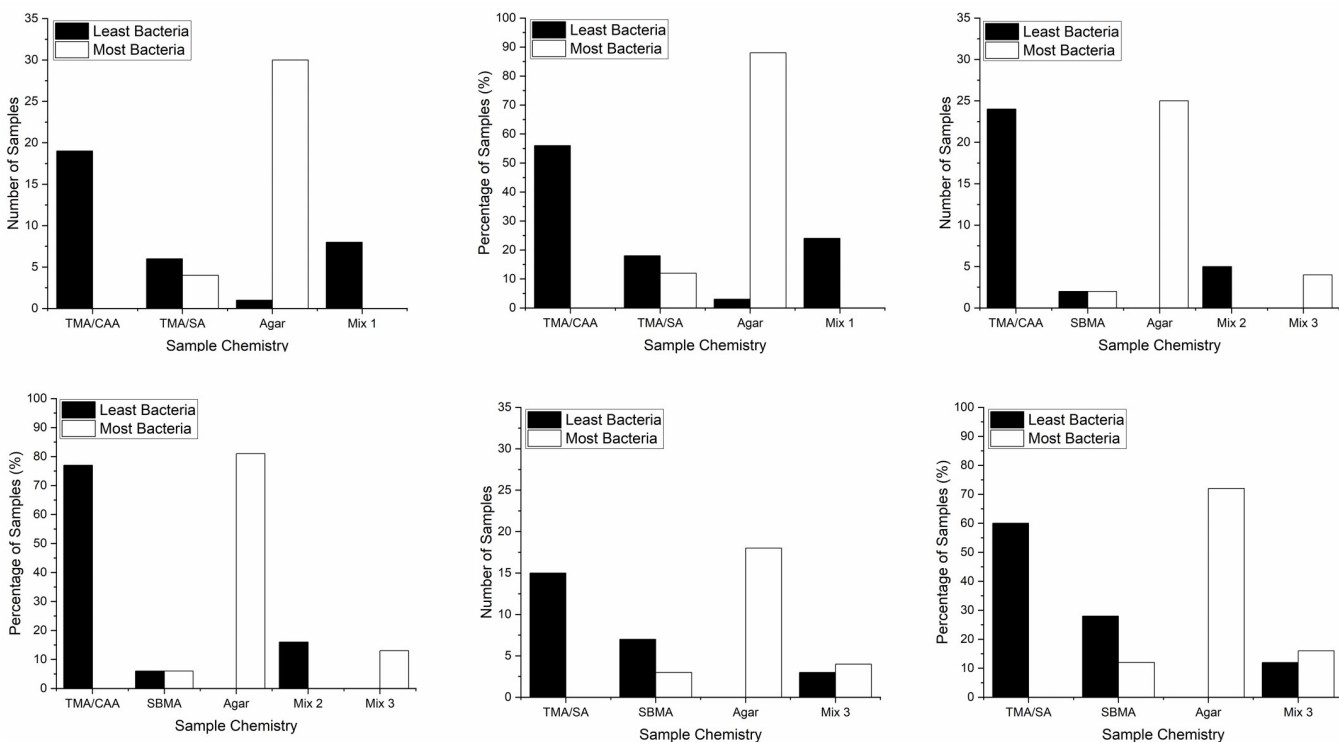

**Fig 3. Citizen scientist results summarized for each experimental combination summarized in Table 1.** Combination 1 (a, b), Combination 2 (c, d), and Combination 3 (e, f) are presented as the total number of samples identified (a, c, and e) or the percentage of each chemistry's samples (b, d, and f). Mix 1 represents results where both TMA/CAA and TMA/SA were identified, Mix 2 represents results where both TMA/CAA and SBMA were identified, and Mix 3 represents results where both SBMA and Agar were identified.

provides better bacteria resistance than the closely related SBMA chemistry. The TMA/SA hydrogel had the least bacteria in 21 samples (35.6%) and the most bacteria in only 4 samples (6.8%), both of which were better than the SBMA results of 9 with the least and 5 with the most (16.1% and 8.9%, respectively). In a subset of the results, multiple samples were identified as having either the least or most bacteria. When multiple chemistries were co-identified as having the least bacteria it was either a combination of TMA/CAA and TMA/SA (Mix 1, 23.5% of the samples) or TMA/CAA and SBMA (Mix 2, 16.1% of the samples). Conversely, all of the multiple chemistry results identified as having the most bacteria were a combination of the SBMA and agar control samples (Mix 3, 14.3% of the samples), further supporting the conclusion that the SBMA hydrogels have the lowest bacteria resistance amongst the nonfouling chemistries. Finally, it is also clear that the agar control performed as expected, promoting the greatest bacteria adhesion levels in 73 of the returned data sets (81.1%), not including the 8 samples where it was co-ranked with SBMA as having the most bacteria.

## Conclusion

In this work citizen scientists in grades 3–5 were used to evaluate the performance of three different nonfouling polymers for their relative bacteria resistance. Randomized hydrogel samples were provided to the students, who exposed them to high contact surfaces in their homes. Bacteria growth was fostered with nutrient broth and repeat bacteria exposure occurred weekly over four weeks. Samples were ranked as having the least or most bacteria coverage after 30 days of experimentation based on visual observations of surface coverage. The results clearly

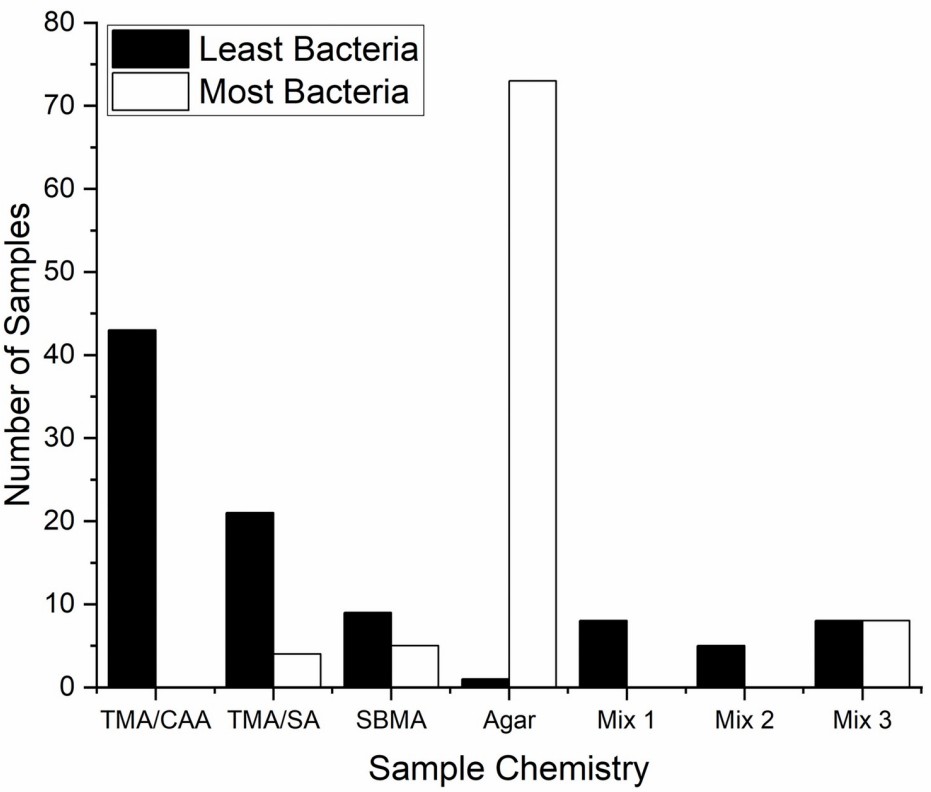

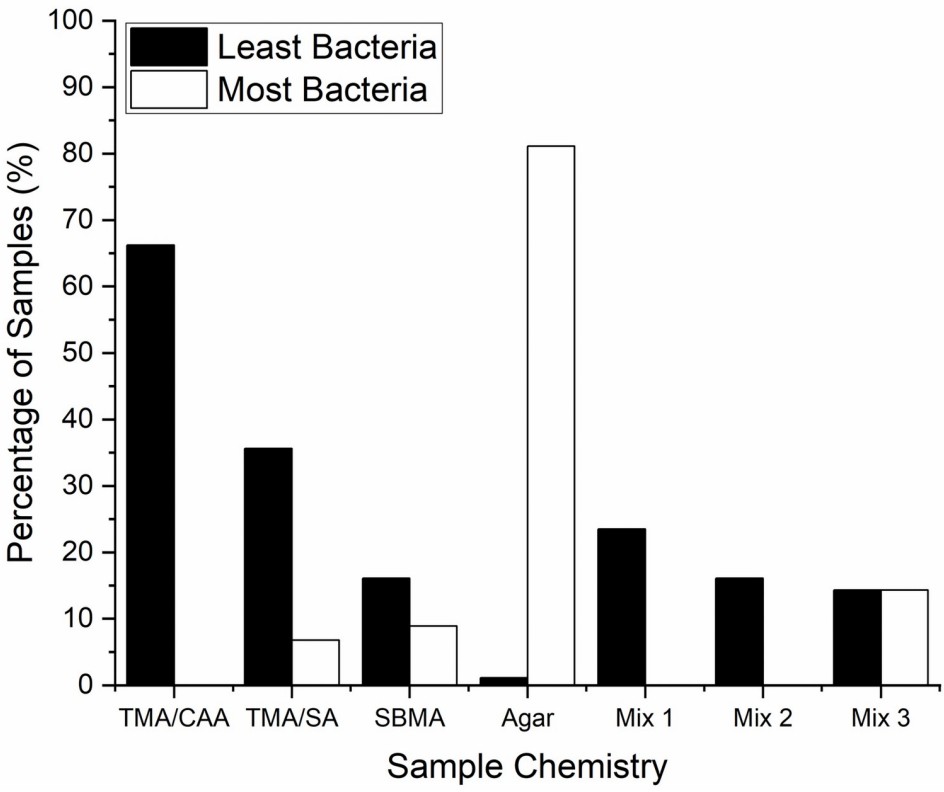

**Fig 4.** Citizen scientist results summarizing the relative performance of the three nonfouling chemistries based on a) total number of samples identified and b) percentage of each chemistry's samples.

indicate that the TMA/CAA polyampholyte chemistry outperformed the other nonfouling chemistries, suggesting that it provides the best long-term resistance to bacteria adhesion.

## Experimental procedure

All chemicals were purchased from Sigma-Aldrich and used as received. All three nonfouling polymer hydrogels were prepared via large polymer sheets in 8 in x 8 in glass baking pans using a scaled-up version of previously published procedures [17, 21, 22]. Each baking pan required ~30 mL of the monomer reaction solution which consisted of 0.2 mol total monomers, a triethylene glycol dimethacrylate cross-linker (TEGDMA), a solvent composed of a 1.5:1.5:1 mixture of 6 M NaOH: ethylene glycol: ethanol, and ammonium persulfate and sodium metabi-sulfite polymerization reaction initiators. The total monomer to cross-linker ratio was fixed at 26.7:1 for all hydrogel chemistries. After filling, the baking pans were placed into an oven at 60˚C for 90–120 minutes. After polymerization, the polymer sheets were removed from the pan and soaked in 0.075 M NaCl for 24 hours, and then stamped into 60 mm diameter polystyrene petri dishes. Each petri dish received 5 mL of 0.075 M NaCl to keep the hydrogels hydrated and they were sealed with parafilm and labeled for the citizen science kits.

The bacteriological agar was produced by mixing 7.5 g agar in 500 mL of DI water in batches. The mixture was heated to 85˚C while stirring until the agar was completely dissolved and then it was removed from the heat source. Once the agar solution cooled to roughly 50˚C, it was transferred into petri dishes in 10 mL aliquots. To avoid contamination from the air, the lid of each petri dish was only removed just prior to charging with solution, and promptly replaced. After about 15 minutes in ambient temperature, the agar began to solidify at which point the petri dishes were moved into a refrigerator. In the refrigerator the petri dishes were place upside down to collect any sample condensate.

The Luria-Bertani broth was prepared by dissolving 75 g of Luria-Bertani powder into 3 L of water. After pouring into glass jars, the broth was autoclaved for sterilization. The sterile broth was then divided into aliquots for the Citizen Science kits.

Each citizen scientist was provided with an experimental kit at the start of their effort. Each kit contained three sample petri dishes, including two with polymer hydrogels and one agar control dish. The petri dishes were labeled with numbers 1–4 to keep the sample identities unknown to the citizen scientists to prevent sample bias. In addition to the hydrogel and agar samples, each kit also contained four 10 mL vials of sterile Luria-Bertani nutrient broth, one syringe for transferring media, twelve sterile cotton swabs for bacteria exposure, a magnifying glass, and an experimental logbook. A total of 200 experimental kits were prepared for the citizen scientists. The citizen scientists self-selected the high contact surface area for their individual experiments, and surfaces included door handles, stairs, bathroom sinks, trash cans, railings, windows, toilet handles, keyboards, and cell phones, etc. Several additional kits were also set up in the school and photographed regularly, for use by students who made mistakes during their individual experiments.

## Acknowledgments

The authors would like to recognize contributions to the project from Dr. Luella Stelck and her students at Russell Elementary School for their contributions to the citizen science and Dr. Stephanie Haag at the University of Idaho for guidance with experimental design.

## Author Contributions

**Conceptualization:** Niko Hansen, Adriana Bryant, Roslyn McCormack, Hannah Johnson, Travis Lindsay, Kael Stelck, Matthew T. Bernards.

**Data curation:** Niko Hansen, Adriana Bryant, Roslyn McCormack, Kael Stelck, Matthew T. Bernards.

**Formal analysis:** Kael Stelck, Matthew T. Bernards.

**Funding acquisition:** Matthew T. Bernards.

**Investigation:** Niko Hansen, Matthew T. Bernards.

**Methodology:** Niko Hansen, Roslyn McCormack, Matthew T. Bernards.

**Project administration:** Matthew T. Bernards.

**Resources:** Matthew T. Bernards.

**Software:** Niko Hansen, Adriana Bryant, Matthew T. Bernards.

**Supervision:** Matthew T. Bernards.

**Validation:** Matthew T. Bernards.

**Visualization:** Matthew T. Bernards.

**Writing – original draft:** Niko Hansen, Adriana Bryant, Roslyn McCormack, Matthew T. Bernards.

**Writing – review & editing:** Niko Hansen, Adriana Bryant, Roslyn McCormack, Hannah Johnson, Travis Lindsay, Matthew T. Bernards.

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
