## [Decision Letter · Decision Letter 0]

14 Sep 2021

PONE-D-21-25045Assessment of the Performance of Nonfouling Polymer Hydrogels Utilizing Citizen ScientistsPLOS ONE

Dear Dr. Bernards,

Thank you for submitting your manuscript to PLOS ONE. After careful consideration, we feel that it is a well designed and well conducted study but requires a few minor edits in order to fully meet PLOS ONE’s publication criteria as it currently stands. Therefore, we invite you to submit a revised version of the manuscript that addresses the points raised during the review process.

We look forward to receiving your revised manuscript.

Kind regards,

Robert Chapman, Ph.D.

Academic Editor

PLOS ONE

Journal Requirements:

2. Acknowledgments Section: Move New Information to the Financial Disclosure:

"Thank you for stating the following in the Acknowledgments Section of your manuscript: 

[This research was funded by the National Aeronautics and Space Administration (NASA) as part of the Student Payload Opportunity with Citizen Science (SPOCS) program under Federal Award No 80NSSC17M0021 as a subaward to the University of Idaho. The authors would like to recognize contributions to the project from Dr. Luella Stelck and her students at Russell Elementary School for their contributions to the citizen science and Dr. Stephanie Haag at the University of Idaho for guidance with experimental design.]

 [This research was funded by the National Aeronautics and Space Administration (NASA) as part of the Student Payload Opportunity with Citizen Science (SPOCS) program under Federal Award No 80NSSC17M0021, as a subaward to the University of Idaho. The funders had no role in study design, data collection and analysis, decision to publish, or preparation of the manuscript.]

3. We note that Figure 1 in your submission contain copyrighted images. All PLOS content is published under the Creative Commons Attribution License (CC BY 4.0), which means that the manuscript, images, and Supporting Information files will be freely available online, and any third party is permitted to access, download, copy, distribute, and use these materials in any way, even commercially, with proper attribution. For more information, see our copyright guidelines: http://journals.plos.org/plosone/s/licenses-and-copyright.

a) You may seek permission from the original copyright holder of Figure 1 to publish the content specifically under the CC BY 4.0 license. 

Academic Editor comments:

The study is a well designed and interesting comparison of three common hydrogels as antibacterial surfaces, using data from a large number of students (~100). The research presented is original and to my knowledge not published elsewhere. The experimental methodology is described in good detail, although a breakdown of the individual experimental combinations would improve the clarity of the manuscript (see reviewer 1 below). The conclusions are clear, and the manuscript is well written. I would recommend publication after a few minor alterations suggested by reviewer 1, including a description of how a sample is determined to have the most bacteria, and a break-down of the individual experimental combinations.

Reviewers' comments:

Reviewer's Responses to Questions

**Comments to the Author**

1. Is the manuscript technically sound, and do the data support the conclusions?

Reviewer #1: Partly

2. Has the statistical analysis been performed appropriately and rigorously? 

Reviewer #1: No

3. Have the authors made all data underlying the findings in their manuscript fully available?

Reviewer #1: Yes

4. Is the manuscript presented in an intelligible fashion and written in standard English?

Reviewer #1: Yes

5. Review Comments to the Author

Reviewer #1: Bernards and coworkers present an investigation into the antifouling behavior of hydrogels prepared from commercially available monomers. The investigation was performed using Citizen Science and showed that materials made from combinations of TMA/CAA had the highest antifouling behavior. The paper is well written and quite intersting, however, I have some concerns about some of the scientific validity that should be addressed before publication.

Major concerns:

1) While the methods are clear, the authors have not explained how a sample was deemed to have the most or least bacteria. I assume this is determined by some visual observation, but this needs to be clearly stated either in the main text or in the experimental section. Was the most bacteria determined by the number of distinct colonies, by the sample that showed the single largest colony, by the sample that showed the highest overall cloudiness, etc.?

2) It was clear that the TMA/CAA hydrogel provided the highest antifouling activity overall, however, the data for the individual combinations should also be reported. For instance, in combination 1 (TMA/CAA, TMA/SA, Agar Control), what is the percentage of observations that showed TMA/CAA showed the least bacteria? I am curious to see whether the overall results are skewed by a disproportionately high/low number of observations in one of the combination sets. For instance, did the TMA/CAA show far superior antifouling behavior vs TMA/SA (100% of observations showing TMA/CAA had the least bacteria) but only modest antifouling behavior vs SBMA?

3) The "multiple" observation in figure 3a is unclear. What happened to this data set when the data was converted to a percentage in figure 3b?

Minor:

Please define LB (Luria-Bertani) in the experimental procedure section.

Other notes:

A control sample made from another commercial monomer may have provided a good comparison with the antifouling hydrogels. The antifouling behavior of the TMA/CAA hydrogels may be, in part, due to the presence of residual chemicals during synthesis (e.g. residual unreacted monomers or ethanol). Performing a control experiment with a standard commercial monomer that exhibits little antifouling behavior (e.g. ethylene glycol methyl ether methacrylate) may have more conclusively shown that the enhanced antifouling behavior compared to agar was not due to the synthetic procedure. This will not affect the present study as the comparison was between the 3 hydrogels made via the same procedure.

6. PLOS authors have the option to publish the peer review history of their article (what does this mean?). If published, this will include your full peer review and any attached files.

Reviewer #1: No

---

## [Author Response · Author response to Decision Letter 0]

15 Nov 2021

A detailed response to the reviewer comments is provided in the cover letter for this resubmission.

---

## [Editor Report · Decision Letter 1]

13 Dec 2021

Assessment of the performance of nonfouling polymer hydrogels utilizing citizen scientists

PONE-D-21-25045R1

Dear Dr. Bernards,

We’re pleased to inform you that your manuscript has been judged scientifically suitable for publication and will be formally accepted for publication once it meets all outstanding technical requirements.

Kind regards,

Robert Chapman, Ph.D.

Academic Editor

PLOS ONE

Additional Editor Comments (optional):

Thank you for your resubmission. The revision has answered all questions raised by the reviewers.
---

## [Editor Report · Acceptance letter]

21 Dec 2021

PONE-D-21-25045R1 

Assessment of the performance of nonfouling polymer hydrogels utilizing citizen scientists 

Dear Dr. Bernards:

I'm pleased to inform you that your manuscript has been deemed suitable for publication in PLOS ONE. Congratulations! Your manuscript is now with our production department. 

Kind regards, 

on behalf of

Dr. Robert Chapman 

Academic Editor

PLOS ONE